# Inhibition of GCN2 Reveals Synergy with Cell-Cycle Regulation and Proteostasis

**DOI:** 10.3390/metabo13101064

**Published:** 2023-10-09

**Authors:** Gregory Gauthier-Coles, Farid Rahimi, Angelika Bröer, Stefan Bröer

**Affiliations:** 1Research School of Biology, Australian National University, Canberra, ACT 2601, Australia; gregory.gauthier-coles@yale.edu (G.G.-C.); farid.rahimi@anu.edu.au (F.R.); angelika.broer@anu.edu.au (A.B.); 2School of Medicine, Yale University, New Haven, CT 06504, USA

**Keywords:** integrated stress response, amino acid, PERK, cyclin-dependent kinases, cell cycle, amino acid transport

## Abstract

The integrated stress response is a signaling network comprising four branches, each sensing different cellular stressors, converging on the phosphorylation of eIF2α to downregulate global translation and initiate recovery. One of these branches includes GCN2, which senses cellular amino acid insufficiency and participates in maintaining amino acid homeostasis. Previous studies have shown that GCN2 is a viable cancer target when amino acid stress is induced by inhibiting an additional target. In this light, we screened numerous drugs for their potential to synergize with the GCN2 inhibitor TAP20. The drug sensitivity of six cancer cell lines to a panel of 25 compounds was assessed. Each compound was then combined with TAP20 at concentrations below their IC_50_, and the impact on cell growth was evaluated. The strongly synergistic combinations were further characterized using synergy analyses and matrix-dependent invasion assays. Inhibitors of proteostasis and the MEK–ERK pathway, as well as the pan-CDK inhibitors, flavopiridol, and seliciclib, were potently synergistic with TAP20 in two cell lines. Among their common CDK targets was CDK7, which was more selectively targeted by THZ-1 and synergized with TAP20. Moreover, these combinations were partially synergistic when assessed using matrix-dependent invasion assays. However, TAP20 alone was sufficient to restrict invasion at concentrations well below its growth-inhibitory IC_50_. We conclude that GCN2 inhibition can be further explored in vivo as a cancer target.

## 1. Introduction

The integrated stress response (ISR) pathway allows cells to adapt to stressful situations, namely, amino acid depletion, endoplasmic reticulum (ER) stress, viral infection, and heme depletion [1]. The ISR includes four protein kinases that respond to these stressors by targeting the eukaryotic translation initiation factor eIF2α. More specifically, general control nonderepressible 2 (GCN2) is a stress-sensing kinase that responds to amino acid depletion using uncharged tRNAs as a sensor of amino acid imbalance; protein kinase R-like endoplasmic reticulum kinase (PERK) senses ER-stress [2]; protein kinase R (PKR) detects double-stranded RNA [3]; heme-regulated inhibitor (HRI) responds to stresses associated with hemoglobin production [4]. In the context of this study, we focused on GCN2.

Phosphorylation of eIF2α reduces overall protein biosynthesis and concurrently allows the translation of specific mRNAs with upstream open-reading frames, particularly ATF4 [5]. ATF4 is a general activating transcription factor that promotes the synthesis of genes involved in amino acid biosynthesis, transport, autophagy, and antioxidant responses [6,7]. These mechanisms can restore amino acid homeostasis by improving amino acid provision and enhancing the recycling of proteins and cellular materials. In cancer cells, rapid growth causes amino acid depletion and ER stress [8], drawing attention to the role of GCN2 and PERK in cancerous cell growth [9,10]. In vitro, however, cancer cell lines are often unstressed and require inhibition of other components of amino acid homeostasis to elicit the ISR [11,12,13]. Indeed, elements of amino acid homeostasis are frequently explored for cancer therapy. Asparaginase, for instance, is used to treat acute lymphocytic leukemia [14]. Meanwhile, inhibition of amino acid transporter LAT1 [15] has been explored to treat various solid tumors. Bortezomib is a proteasome inhibitor that affects protein recycling [16], while rapamycin-derived mTORC1 inhibitors have been evaluated in numerous clinical trials [17].

Activation of GCN2 initiates a rescue mechanism for cancer cells to respond to cellular stresses elicited by anticancer drugs, thus thwarting the intended effects of perturbing amino acid homeostasis [13,16,17]. Thus, pairing a drug that targets amino acid homeostasis with a GCN2 inhibitor could improve the therapeutic efficacy. Combining cancer drugs can suppress drug resistance and achieve better outcomes [18]. Monotherapies selectively affect a heterogenous population of malignant cells, often resulting in drug resistance in recurring cancers [19]. One strategy to target diverse cell populations is combination therapy. Several large screening studies have identified promising drug combinations, including clofarabine with bortezomib, nilotinib with paclitaxel [20], and irinotecan and CHEK1 inhibitors [21].

Several GCN2 inhibitors have been identified and characterized for kinase selectivity and pharmacological properties. Nakamura et al. [14] reported the discovery of GCN2iB with an IC_50_ of 2.4 nM. GCN2iB also inhibits other kinases, including MAP2K5, STK10, and ZAK, with slightly lower potencies. Nevertheless, it has a suitable pharmacokinetic profile for in vivo studies. GZD824 (olverembatinib) was initially reported as a Bcr-Abl inhibitor and was recently found to inhibit GCN2; however, it shows low specificity in kinase assays [22]. Lastly, triazolo [4,5-d]pyrimidines (TAP) are a class of GCN2 inhibitors that were developed further by Dorsch et al., [23,24,25]. Among them, compound 20 (TAP20) exhibits high potency and selectivity. In vitro, TAP20 inhibits GCN2 with an IC_50_ of 17 nM; it is similarly potent against GSK3α/β and, to a lesser extent, CDK9/cyclinD1.

GCN2 is activated in cancers that experience stress due to nutrient depletion in the tumor environment, elevated protein biosynthesis, and limited vascularization [26]. Activation of GCN2 allows tumor cells to adapt to changes in their microenvironment and promotes quiescence, causing therapy resistance [27]. Thus, combining conventional cancer therapy with concomitant inhibition of GCN2 could overcome therapeutic resistance and improve treatment outcomes.

In this study we explore the synergistic relationships between 25 experimental and approved drugs and the GCN2 inhibitor TAP20 using several cell lines.

## 2. Materials and Methods

Cell lines: All cell lines were procured from the American Type Culture Collection (ATCC) and maintained in DMEM/F-12 with 10% heat-inactivated FBS, 2 mM glutamine, and penicillin (10 U/mL)/streptomycin (10 μg/mL). Cells were incubated in a humidified atmosphere with 5% CO_2_ at 37 °C in 25 cm^2^ or 75 cm^2^ culture flasks. Once confluent, cells were trypsinized using 0.25% Trypsin-EDTA (Gibco). Experiments were conducted using cells passaged 5–20 times.

Chemicals and antibodies: Drugs and chemicals were purchased from several sources. TAP20 was synthesized and provided by Merck KGaA (Darmstadt, Germany). YH16899 was purchased from AOBIOUS (Gloucester, MA, USA), thapsigargin from Thermo Fisher Scientific (Waltham, MA, USA), Bay876 from Sigma-Aldrich (St. Louis, MO, USA), and all other compounds from MedChemExpress (Monmouth Junction, NJ, USA). Bumetanide and 5-fluorouracil stocks were dissolved in water, and all other compounds were dissolved in DMSO. Cetuximab was purchased from MedChemExpress, and all other antibodies were from Cell Signaling Technology (Danvers, MA, USA).

Proliferation assay: Cell growth was monitored using the IncuCyte system (Essen Biosciences). Cell confluence was measured only at the end of the assay to increase throughput. Briefly, 3000 cells were seeded per well in 96-well flat-bottom plates in a growth medium and allowed to adhere overnight. The next day, the media was replaced with 300 µL of growth medium per well with or without treatment. Cells were incubated (5% CO_2_ humidified atmosphere; 37 °C) for a predefined period before being scanned by the IncuCyte system. This period ranged from four to ten days and was determined based on the shortest time required for the untreated control cells to reach >90% confluence. Confluence was recorded as the mean from two images taken for each well. Peripheral wells were excluded from the analysis.

IC_50_ curves were plotted from data obtained in this assay using the following logistic growth equation:y=A1−A21+(xx0)p+A2
where *A*_1_ is the maximal confluence, *A*_2_ is the minimal confluence, *x* is the inhibitor concentration, *x*_0_ is the inhibitor concentration at half-maximal effect, and *p* is the power value.

To evaluate synergism among drug pairings, the coefficient of drug interaction [28] (CDI) was calculated using the following equation:$$CDI=\frac{AB}{A\times B}$$
where *AB* is the confluence of cells treated with the drug combination, and *A* and *B* are the confluence values of cells exposed to each drug alone. CDI scores of 1 indicate an additive effect, those >1 indicate antagonism, and those <1 indicate synergism.

Synergy analysis: Synergy diagrams were generated from data produced in the assay described above. A matrix of seven concentrations per drug was arrayed using MDA-MB-231 and HPAF-II cells seeded across three 96-well plates for each experiment. Drug concentrations were incremented by values of 1, 2, and 4 per order of magnitude. Peripheral wells were excluded from analysis, and each plate contained untreated controls to which the confluence values of treated cells were normalized. Normalized data were then converted to synergy scores using SynergyFinder [28] and the Loewe equation.

Western blotting: Cells were grown to near confluence in 35 mm dishes. Proteins were prepared for SDS-PAGE by washing cells with PBS (pH 7.4) and adding 100–150 μL of RIPA lysis and extraction buffer (Thermo Fisher Scientific; Waltham, MA, USA) containing cOmplete Mini Protease Inhibitor Cocktail (Roche; Basel, Switzerland). The lysate was collected using a cell scraper and transferred to a microcentrifuge tube. After 5 min of mixing and 5 min of centrifugation at top speed, 10 μL of the supernatant was sampled and diluted at least ten-fold in water for the Bradford assay (Sigma-Aldrich; St. Louis, MO, USA) to measure protein concentration and normalize the protein mass added to the wells of an SDS-PAGE gel. Lysates were mixed with 4× Bolt LDS, 10× Bolt Sample Reducing Agent (Life Technologies), and water up to 50 µL. A volume of the mixture, containing 20–30 µg protein, was then loaded into the wells of a precast NuPAGE 4–12% Bis-Tris 10- or 15-well gel fitted to a Mini Gel Tank (Life Technologies) filled with MOPS SDS running buffer (Thermo Fisher; Waltham, MA, USA). Sample lanes were flanked with wells loaded with SeeBlue Plus2 or Novex Sharp pre-stained protein standards (Invitrogen; Waltham, MA, USA). Proteins were separated for approximately 1 h at 150 V and transferred onto a 0.45-µm Amersham Protran nitrocellulose membrane (GE Healthcare; Chicago, IL, USA) in an ice-cold wet transfer system (Bio-Rad; Hercules, CA, USA) containing Towbin buffer (192 mM NaCl, 25 mM Tris base, 20% methanol (*v*/*v*); pH 8.3) for 2 h at 100 V. The blot was then incubated in 5% (*w*/*v*) skim milk powder in PBS-Tween (0.1% Tween-20; pH 7.4) for 1 h at room temperature or overnight at 4 °C. Between blocking and antibody treatment, blots were washed thrice with PBS-Tween for 5 min at a time. Primary antibodies were incubated with blots overnight, while secondary antibodies were incubated for 2–6 h. To visualize the proteins, either the SuperSignal West Pico PLUS or the SuperSignal West Femto Maximum Sensitivity chemiluminescent substrates (Thermo Scientific; Waltham, MA, USA) were added dropwise to the blots. The ChemiDoc MP (Bio-Rad) or Fusion Solo 2 (Vilber) imaging systems were used to visualize chemiluminescence.

All blots were probed first for the protein of interest and then for a housekeeping protein. β-Actin was probed as the loading control. To remove protein-bound antibodies from the previous immunodetection, blots were incubated at 65 °C in stripping buffer (2% SDS (*w*/*v*), 62.5 mM Tris, and 0.7% 2-mercaptoethanol (*v*/*v*); pH 6.8) for 10 min with agitation. The blots were then washed with PBS-Tween and incubated for at least 2 h with 5% skim milk in PBS-Tween.

Matrix-dependent invasion assay: To investigate drug-pair effects on matrix-dependent invasion, 143B osteosarcoma cells were used in a Cultrex 3D spheroid invasion system (R&D Systems; Minneapolis, MN, USA) as they readily form the necessary spheroids, unlike the six cell lines used in the synergy screening. Initially, 143B were seeded in 1 × spheroid formation medium (ECM; 3000 cells/well in 50 μL) into low-adhesion, round-bottom 96-well plates to induce spheroid formation. The plates were centrifuged at 300× *g* for 3 min to pellet the cells. Working on ice, 50 μL of invasion matrix was added to the spheroids after 72 h. Subsequently, plates were centrifuged at 300× *g* and 4 °C to precipitate the spheroids. Gel formation was initiated by incubation at 37 °C for 1 h, followed by the addition of 100 μL of pre-warmed culture medium. Plates were then incubated at 37 °C/5% CO_2_ in a humidified incubator for 3–6 days, and images were captured with a Leica M205FA stereomicroscope. The invasion perimeter area was measured using the Leica Application Suite, version 4.12.

## 3. Results

We hypothesized that current cancer drugs and experimental anticancer compounds induce stress in cell lines, which is counteracted by GCN2 activation. Thus, blocking GCN2 with TAP20 might reduce the cellular response flexibility, decreasing cellular fitness.

We selected 24 experimental compounds with well-known targets (Table 1) and potential or reported association with the GCN2–ATF4 axis. Halofuginone and YH16899 target tRNA synthetases, which are directly involved in activating GCN2 [29,30]. CB5083 and Bortezomib inhibit protein recycling, an essential aspect of amino acid homeostasis [31,32]. V-9302, NCT-503, and CB-839 interfere with amino acid transport and metabolism, two elements of amino acid homeostasis [33,34]. AMG PERK44 and thapsigargin are linked to ATF4 expression through endoplasmic reticulum stress [35,36]. Bay-876, GSK2837808A (LDH-A inhibitor), and AZD3965 (MCT1 inhibitor) were selected based on a reported association between glucose availability and the ISR [37]. Moreover, BRAF inhibitors activate the ISR [38], while cetuximab—an EGFR inhibitor—downregulates the amino acid transporter SLC1A5, an essential element of amino acid homeostasis [13]. Cell cycle inhibitors danusertib, flavopiridol, seliciclib, and THZ-1 were included due to their relationship with ISR [39]. Similarly, the MEK–ERK pathway was targeted with SCH772984 and selumetinib [40]. Meanwhile, 5-fluorouracil treatment is associated with the formation of stress granules, which are related to the ISR [41]. Bumetanide was selected for its effect on cellular volume regulation, a stress not covered by the ISR [42]. Agerafenib is an inhibitor of BRAF and RET kinase, having an antagonistic relationship with ATF4 [43]. The reported target IC_50_ values for all compounds are in the nanomolar range, typical for modern pharmaceutical compounds. To establish baselines for potential synergistic effects with TAP20, we determined the IC_50_ of the 24 candidate drugs (Table 2) for cell growth using various cell lines with different oncogenic signatures and tissue origin [44]. Growth IC_50_ values can be higher than the target IC_50_ values due to membrane permeability, drug-resistance mechanisms, and cellular reliance on the target function.

To determine synergy with TAP20, candidate drugs were adjusted to a concentration where inhibition was just observable (i.e., <IC_50_, Table 3). In this range, synergy is readily observed as growth rates are sensitive to reduced fitness. Generally, we did not employ concentrations >10 µM due to insolubility or increased nonspecific drug interactions. The dose-response curves for all cell lines are presented in Appendix A.

We also determined the response of cell lines to TAP20. TAP20 alone typically does not inhibit growth when cell lines are cultured in replete growth media as they experience minimal stress that would initiate activation of GCN2 (Appendix A). To quantify the action of TAP20 on GCN2, we starved the cell lines for 24 h in the presence of different TAP20 concentrations (Figure 1, densitometry in Appendix A). The expression of ATF4 was the most reliable indicator of GCN2 activation; thus, we used western blotting to determine ATF4 expression at different TAP20 concentrations. Except for Panc 02.03 cells, ATF4 expression was suppressed by TAP20 concentrations >1 µM. The IC_50_-values were as follows: MDA-MB-231 = 0.54 µM, HPAFII = 0.56 µM, and SKOV3 = 0.29 µM. TAP20 was used at 3 µM for all synergy tests to ensure that the ISR was not activated via GCN2.

In the synergy experiments, control cells were grown to confluency, and the growth was compared to that of the treated cells. A coefficient of drug interaction was calculated and used to compare the results (Table 4). A score < 1 (green in Table 4) indicates a synergistic relationship, a score of 1 indicates an additive effect (yellow), and scores > 1 indicate an antagonistic relationship (red). The unprocessed growth data for each combination are shown in Appendix A. Several trends were observed. First, instances of drug synergy were not consistently observed across all cell lines. We observed various synergistic relationships in MDA-MB-231, HPAFII, and SKOV3 cells, which were primarily additive in MDA-MB-468 and Panc02.03 cells. Meanwhile, we only observed a limited number of synergistic relationships in Ovcar3 cells. Second, rapamycin was antagonistic to TAP20. This was expected as inhibition of mTORC1 activates GCN2 and enhances eIF2α phosphorylation [45].

Numerous synergistic relationships between TAP20 and the candidate drugs were identified and related to several cellular processes. For instance, V-9302 inhibits amino acid transporters LAT1 and SNAT2 [33]. Both transporters play crucial roles in maintaining cytosolic amino acid levels. Inhibition of LAT1 has been shown to activate GCN2 [46]. Similarly, we expected inhibitors of amino acid homeostasis to synergize with TAP20. Halofuginone and YH16889 interfere with prolyl-tRNA synthetase and lysyl-tRNA synthetase, respectively, and are mechanistically related to the tRNA-binding function of GCN2. CB-5083 inhibits p97 ATPase, a protein involved in unfolding ubiquitinated proteins and membrane proteins [47]. The unfolded proteins are then transferred to the proteasome for degradation, replenishing cellular amino acid pools. Another inhibitor of proteostasis is bortezomib, which blocks the proteasome, although synergy was observed in different cell lines compared to CB-5083. Thapsigargin, an inhibitor of the SERCA ATPase, induces ER stress, an independent activator of GCN2 through the protein kinase PERK. However, thapsigargin only synergized with TAP20 in HPAFII and SKOV3 cells.

A direct relationship between the MEK–ERK MAPK signaling pathway and ISR has been demonstrated in HepG2 cells [40]. Consistently, we found a strong synergy between TAP20 and the ERK inhibitor SCH772984. The MEK inhibitor selumetinib, which inhibits the MEK–ERK pathway upstream of ERK, had a similar effect, although not in HPAFII cells.

The most striking synergy we observed was between elements of cell-cycle regulation and TAP20. This was exemplified by inhibitors of cyclin-dependent kinases, such as flavopiridol, seliciclib, and THZ-1 [48] and danusertib—an inhibitor of aurora kinases [49]. Although flavopiridol and seliciclib are non-specific CDK inhibitors, THZ-1 is a more selective CDK-7 inhibitor.

We performed growth experiments to evaluate the scope of these synergistic relationships further, exploring a matrix of different drug concentrations and assessing the synergistic effect on matrix-dependent cell invasion. Due to the low invasiveness of the cell lines in the growth assays, we performed the cell invasion assay with 143B osteosarcoma cells, a cell line for which we optimized the assay previously [24]. The synergy analysis was performed with MDA-MB-231 and HPAF2 cells, which showed the most robust responses in this study.

Combining TAP20 with the p97-ATPase inhibitor CB5083 showed a synergistic effect, as evidenced by growth inhibition and the synergy scores calculated using Synergyfinder [28]. In HPAFII cells, the optimal effect was achieved at 1 µM CB5083 and >1 µM of TAP20 (Figure 2).

Although TAP20 alone had little effect on growth in all cell lines, it dramatically reduced matrix-dependent invasion of 143B cells at 1–3 µM; however, to evaluate the synergistic effects, a lower concentration, 0.75 µM, was used (Figure 2). In agreement with the growth experiments, spheroid size was reduced in the presence of both inhibitors (compare the growth area of the control (+) with that of TAP20/CB5083).

The highly synergistic relationship between the CDK inhibitor flavopiridol and TAP20 was confirmed in the synergy analysis, yielding scores up to 40 in MDA-MB-231 and HPAFII cells (Figure 3). Optimal synergy was achieved with relatively low concentrations of both compounds (MDA-MB-231 TAP20: 0.2 µM, flavopiridol: 0.04 µM, Figure 3a,b; HPAFII TAP20: 1 µM, flavopiridol: 0.03 µM, Figure 3c,d). Synergy was lost at higher drug concentrations in MDA-MB-231 cells (Figure 3b). As shown in Figure 2, TAP20 (0.75 µM) significantly affected the matrix-dependent invasion of 143B cells alone, similar to the effect of 50 nM flavopiridol (Figure 3e). This suggests that invasion generates cellular stress, which is overcome by activating GCN2. Meanwhile, combining the drugs significantly potentiated the effect. In the presence of both drugs, hardly any cell protrusions from the original spheroid were observed (Figure 3f).

This result was confirmed using the non-specific CDK inhibitor seliciclib (Figure 4). Inhibition of MDA-MB-231 cell growth and synergy markedly increased when both drugs were applied at higher concentrations (Figure 4a,b). Synergy was confined to a narrow range of concentrations. At 10 µM, seliciclib significantly reduced matrix-dependent invasion, however, in combination with TAP20 (1 µM), no outgrowth was observed compared to tumor spheroids in the absence of matrix (Figure 4c,d).

Recently, CDK7 has been implicated in the ISR [39]. Hence, to investigate whether this isoform might underly the responses observed for flavopiridol and seliciclib, we used THZ-1, a specific CDK7 inhibitor. Although THZ-1 exhibited a robust synergistic action in MDA-MB-231 (Figure 5a,b) and HPAFII (Figure 5c,d) cells, it was ineffective alone or in combination with TAP20 to suppress growth or invasion of 143B cells (Figure 5e,f).

We also analyzed the MEK and ERK inhibitors selumetinib (Figure 6) and SCH772984 (Figure 7). Both compounds inhibited growth at low concentrations (Figure 6a and Figure 7a) and showed high synergy scores at low concentrations (SCH772984 0.4 µM, selumetinib 2 µM). However, synergy was generally low for SCH772984, which was cytotoxic independent of TAP20. At 100 nM, selumetinib reduced invasion and the spheroid size (Figure 6c,d), suggesting that cells were dying during the treatment. A similar pattern was observed with SCH772984 (Figure 7). Spheroids were markedly smaller than control spheroids grown in the absence of matrix (Figure 7c,d).

In summary, we have identified critical cellular processes, the inhibition of which causes nutrient stress and requires metabolic flexibility, which is afforded by the activation of GCN2. This can be abrogated by concurrent inhibition of GCN2. Hence, combining drugs is an important strategy to overcome drug resistance and inhibit processes that require significant metabolic and signaling flexibility, such as metastasis [18,50,51].

## 4. Discussion

While cancer cell lines experience minimal nutritional stress in vitro, the inverse is often true for cancer cells in the tumor microenvironment [52,53]. Evidence for this is the close association between cancer stem cells and blood vessels [54]. The ISR is a cellular program that deals with stress to elicit responses that optimize the cellular fitness [1,55]. Recycling amino acids through lysosomal and proteasomal protein degradation is key to maintaining amino acid homeostasis [56]. Consistently, we observed synergy with the proteasomal inhibitor bortezomib, the p97-ATPase inhibitor CB-5083, and the amino acid transport inhibitor V-9302. We did not detect synergy with the PERK inhibitor AMG PERK44, suggesting that our selected cancer cell lines did not experience significant ER stress. However, AMG PERK44 might synergize well with CB-5083 or thapsigargin, which cause ER stress. Notably, ATF4 expression could not be suppressed by TAP20 in Panc 02.03 cells. This suggests that a GCN2-independent mechanism, such as PERK, may have upregulated ATF4 expression. Accordingly, we found limited synergy in our screening with this cell line.

A synergistic relationship between TAP20 and inhibitors of cyclin-dependent kinases may not appear intuitive. However, before undergoing division, cells must ensure that sufficient amino acids are available for abundant protein synthesis to duplicate cellular content. Phosphorylation of eIF2α by GCN2 or PERK inhibits the translation of cyclin D1, leading to cell-cycle arrest at the G1 phase [57]. In addition, amino acid deprivation stops cell-cycle progression by stabilizing the mRNA of endogenous cyclin-dependent kinase inhibitors p22 and p27 [40]. Starved cells withdraw from the cell cycle by a mechanism involving transcription factor p53 [58]. This is mediated by stimulation of the cyclin-dependent kinase 1 inhibitor p21 [59]. Thus, multiple mechanisms connect amino acid homeostasis with cell cycle control.

CDK-7 is an unusual cyclin-dependent kinase, as it appears to activate other CDKs instead of directly regulating cell cycle steps. In addition, it has a role as a transcriptional activator [60] and allows RNA polymerase II to initiate transcription while facilitating mRNA capping [61]. A direct link between CDK-7 and the amino acid transporter SNAT2 expression was identified by Stretton et al. [39]. Upregulation of SNAT2 mRNA in response to amino acid limitation depends on CDK-7 activity. While RNA transcription and execution of cellular programs are generally related, activation of GCN2 silences CAP-dependent transcription, allowing mRNAs with uORFs, such as SNAT2, to be translated by a CAP-independent mechanism [62]. Sirtuin 6 was recently identified as a linkage between CDK-7 and the ISR [63]. Sirtuin 6 binds to and stabilizes ATF4, preventing its degradation. Meanwhile, low sirtuin 6 expression causes cells to become sensitive to CDK7 inhibition. CDK-7 enhances the transcription of more than 2000 genes, including MYC [63]. Thus, inhibiting CDK-7 will cause cellular stress, eliciting the ISR.

Related to cell-cycle progression, Piecyk et al. found synergy between oxaliplatin and GCN2 inhibition in colon cancer tumoroids [64]. This was based on GCN2 sustaining ribosomal RNA transcription in nutrient-rich conditions, which synergized with the inhibition of RNA polymerase I.

Clinical trials with CDK inhibitors, such as flavopiridol, have shown limited efficacy even in combination with other chemotherapeutics [59]. This could be due to GCN2 activation, causing cells to become quiescent. The non-specific CDK inhibitor seliciclib exhibited significant toxicity in clinical studies at doses corresponding to effective concentrations in preclinical models [65]. Notably, inhibitors of proteostasis, such as CB-5083 and bortezomib, significantly influence cell-cycle progression as rapid breakdown of cell-cycle proteins is required at certain stages of the cell cycle [47,66]. This may contribute to the synergy observed in this study.

The observed synergy with the MEK–ERK pathway is not readily explained. In the absence of stress, growth-promoting signals lead to Akt-dependent or ERK-dependent phosphorylation of the tuberous sclerosis-2 (TSC2), causing the release of the TSC complex from the lysosome and mTORC1 activation [67,68]. Thus, we might have expected an antagonistic relationship similar to rapamycin. However, a direct link between the MEK–ERK junctures of the MAPK signaling pathway and the ISR has been demonstrated previously in HepG2 cells [40]. In these cells, an ERK inhibitor prevented increased transcription of ATF4 and of the amino acid response target gene SNAT2 [69]. They further showed that phosphorylation of eIF2α was blocked by ERK inhibition without ERK directly phosphorylating eIF2α. Hence, this interaction is interdependent, as GCN2 downregulation prevents ERK phosphorylation in response to amino acid limitation. Moreover, there is a direct relationship between the activity of the amino acid transporter SNAT2 and ERK1/2 activity [70]. Consequently, blocking the MEK–ERK axis could increase nutrient stress, which can be aggravated by simultaneously blocking GCN2. Clinically, the combined use of the MEK inhibitor selumetinib with cisplatin and gemcitabine in patients with advanced biliary tract cancer has not improved outcomes [71].

In this screening, we found that synergy is strongly cell line-dependent. A recent systematic survey of drug interactions by Jaaks et al. found that only 5.2% of >100,000 combinations were synergistic [21]. Intriguingly, the highest rate of synergy was found in pancreatic cancer. Here, we also identified HPAF-II cells as particularly sensitive to combination treatment. Due to the exocrine epithelial origin of pancreatic adenocarcinomas, protein synthesis is likely high, and therefore, HPAF-II cells might be particularly vulnerable to amino acid limitation. Moreover, the GCN2–ATF4 pathway, as a stress-induced response, is more likely to result in synergistic relationships, potentially explaining the higher incidence of synergism (11%) in our overall panel. The study by Jaaks et al. [21] also reported high synergy with cell cycle inhibitors.

Our results highlight the mutual relationship between nutrient stress and the ability of cancer cells to respond to stressors using the GCN2–ATF4 pathway adaptively. While blocking the GCN2–ATF4 axis alone may exhibit a limited ability to reduce cancer cell growth, its blockade in conjunction with stress-inducing drug regimens could be a promising strategy to reduce cancer cell growth and invasion.

## Figures and Tables

**Figure 1 metabolites-13-01064-f001:**
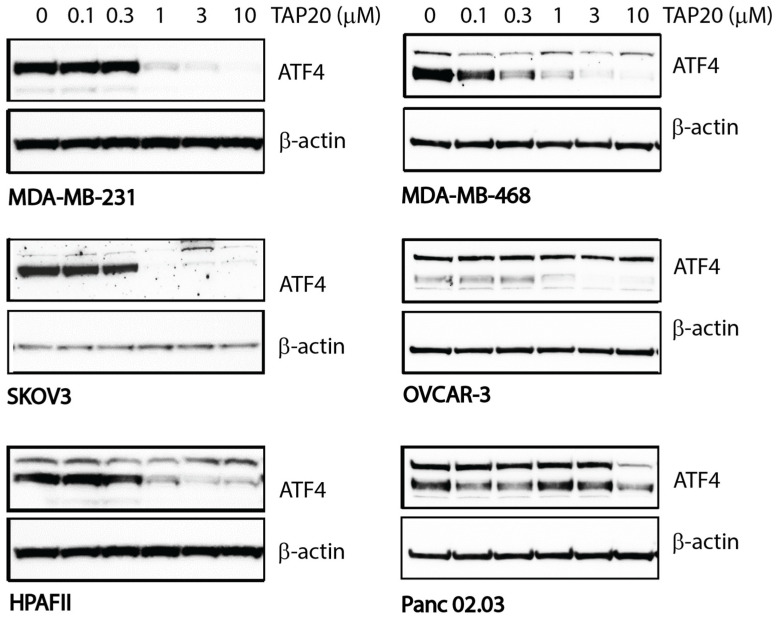
Inhibition of GCN2 by TAP20 in different cell lines as ascertained by ATF4 expression. Cell lines were incubated in amino acid-free media for 24 h in the absence or presence of increasing concentrations of TAP20. ATF4 expression was evaluated by immunoblotting; β-actin was used as the loading control. In some cell lines, ATF4 immunodetection revealed two bands of which the lower band was used for quantification due to its molecular weight. Except Panc 02.03 cells, ATF4 expression was strongly inhibited with >1 µM TAP20.

**Figure 2 metabolites-13-01064-f002:**
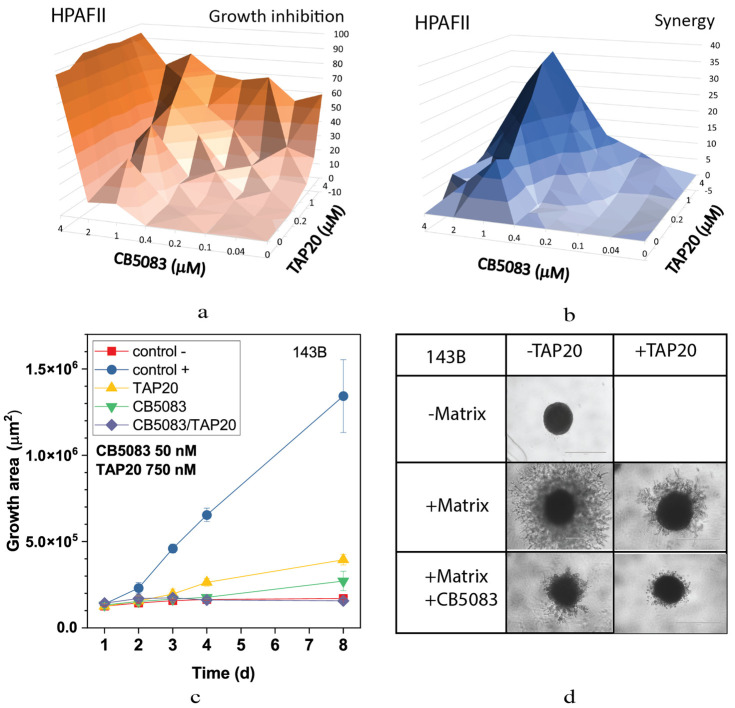
Effect of TAP20 and CB5083 on cell growth and matrix-dependent invasion. For synergy analysis HPAFII cells were grown in the presence of different concentrations of TAP20 and CB5083 (n = 3). Growth is expressed as a percentage of the uninhibited control (**a**). Synergy scores for the experiment are shown in (**b**). (**c**,**d**) 143B osteosarcoma cells were grown to form spheroids before adding the extracellular matrix for invasion. Invasion areas were measured by microscopy (n = 6). (**c**) Quantitative evaluation of outgrowth over eight days. (**d**) Examples of spheroid formation in the presence or absence of inhibitors. Outgrowths with or without matrix are shown as controls.

**Figure 3 metabolites-13-01064-f003:**
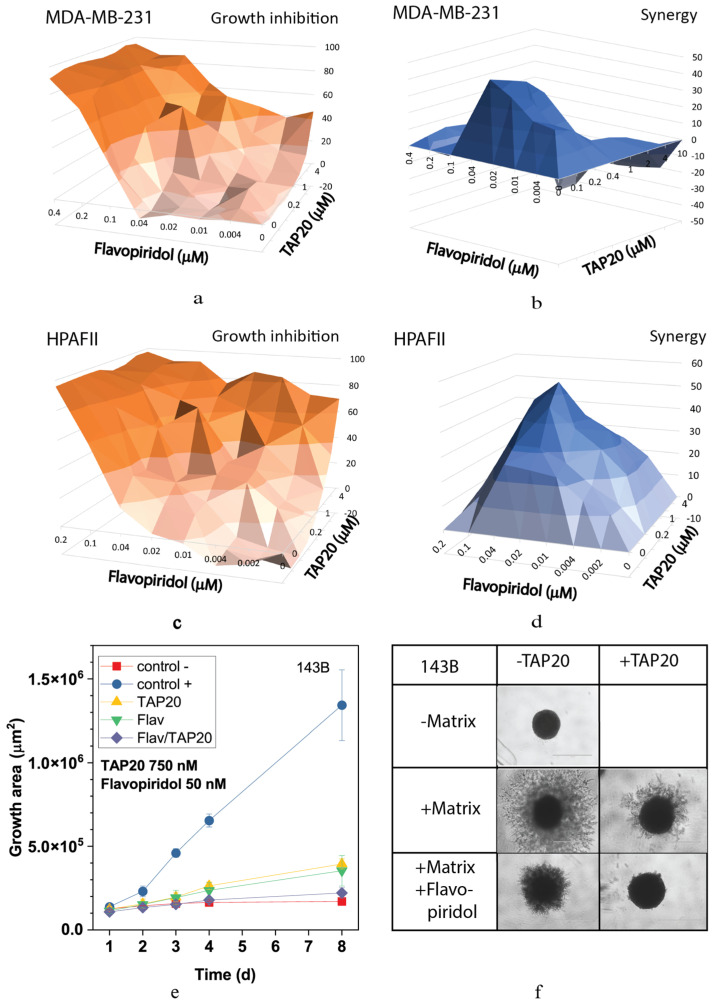
Synergy analysis of TAP20 and flavopiridol and their effects on matrix-dependent invasion. For the synergy analysis, MDA-MB-231 (**a**,**b**) and HPAFII (**c**,**d**) cells were grown in the presence of different combined concentrations of TAP20 and flavopiridol (n = 3). Growth inhibition data (**a**,**c**) were used to calculate synergy scores (**b**,**d**). For matrix-dependent invasion, 143B osteosarcoma cells were grown to form spheroids before adding the extracellular matrix for invasion (n = 6). Invasion areas were measured by microscopy. (**e**) Quantitative evaluation of outgrowth over eight days. (**f**) Examples of the effects of flavopiridol and TAP20 on invasion. Controls are the same as in Figure 2).

**Figure 4 metabolites-13-01064-f004:**
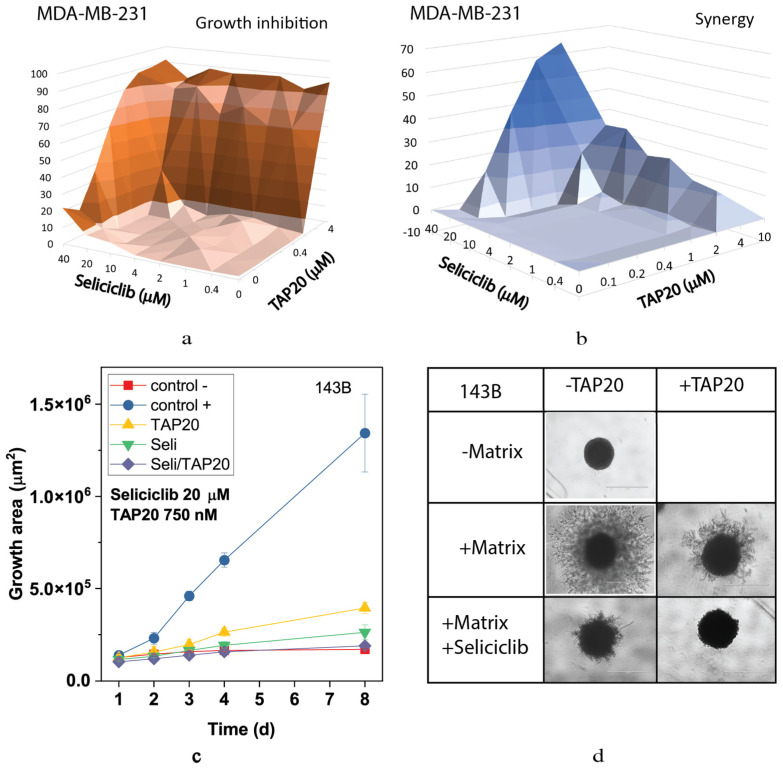
Synergy analysis of TAP20 and seliciclib and their effects on matrix-dependent invasion. For synergy analysis, MDA-MB-231 cells were grown in the presence of different combined concentrations of TAP20 and seliciclib (n = 3). Growth data (**a**) were used to calculate synergy scores (**b**). For matrix-dependent invasion, 143B osteosarcoma cells were grown to form spheroids before the extracellular matrix was added for invasion (n = 6). Invasion areas were measured by microscopy. (**c**) Quantitative evaluation of outgrowth over eight days. (**d**) Examples of the effects of seliciclib and TAP20 on invasion. Controls are the same as in Figure 2.

**Figure 5 metabolites-13-01064-f005:**
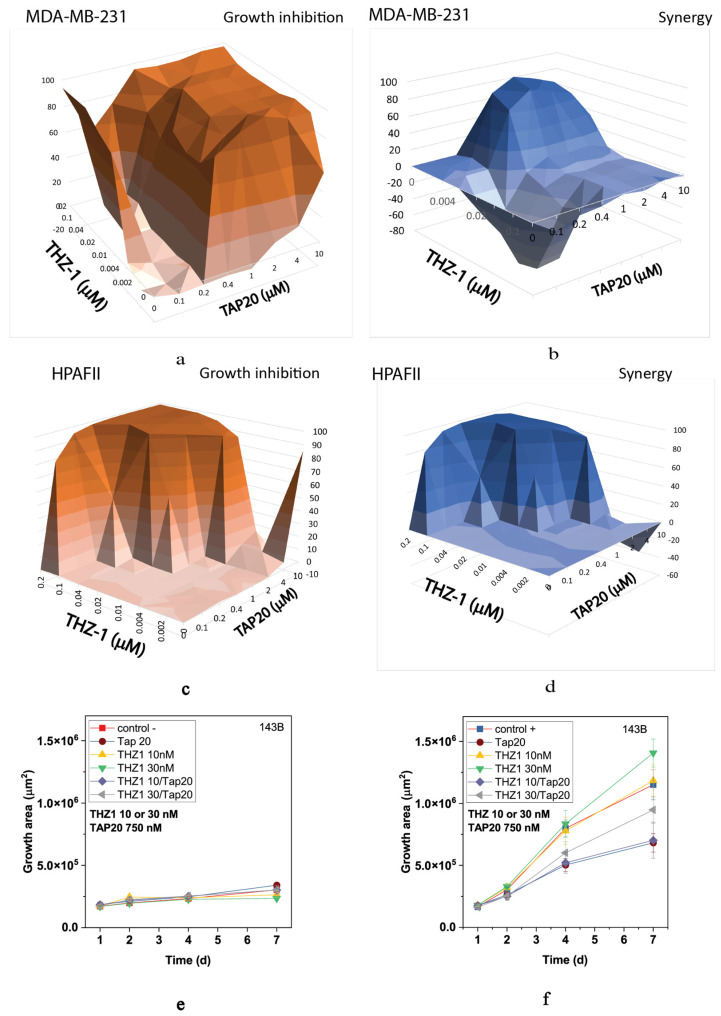
Synergy analysis of TAP20 and THZ-1 and their effect on matrix-dependent invasion. For synergy analysis, MDA-MB-231 (**a**,**b**) and HPAFII (**c**,**d**) cells were grown in the presence of different combined concentrations of TAP20 and THZ-1 (n = 3). For matrix-dependent invasion, 143B osteosarcoma cells were grown to form spheroids before adding the extracellular matrix for invasion (n = 6). Growth and invasion areas were determined by microscopy. (**e**) Growth of spheroids in the absence of matrix and the presence of different drug combinations. (**f**) Outgrowth into the matrix over seven days in the presence of the same drug combinations.

**Figure 6 metabolites-13-01064-f006:**
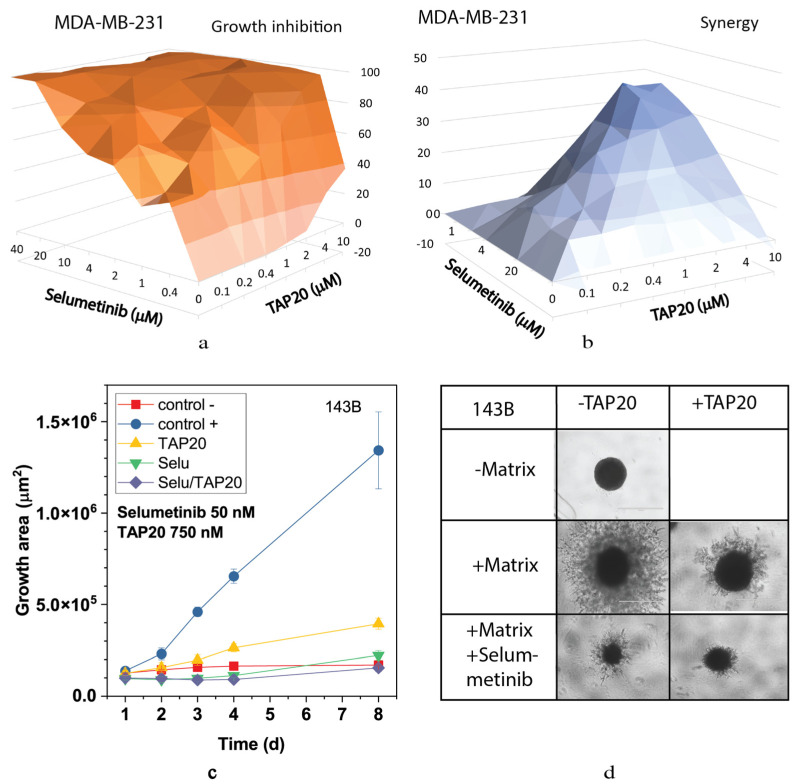
Synergy analysis of TAP20 and selumetinib and their effects on matrix-dependent invasion. For synergy analysis, MDA-MB-231 cells were grown in the presence of different combined concentrations of TAP20 and selumetinib (n = 3). (**a**) Growth inhibition and (**b**) synergy score for different combinations of selumetinib and TAP20. For matrix-dependent invasion, 143B osteosarcoma cells were grown to form spheroids before adding the extracellular matrix for invasion (n = 6). Invasion areas were measured by microscopy. (**c**) Quantitative evaluation of outgrowth. (**d**) Example of the effects selumetinib and TAP20 on invasion. Controls are the same as in Figure 2.

**Figure 7 metabolites-13-01064-f007:**
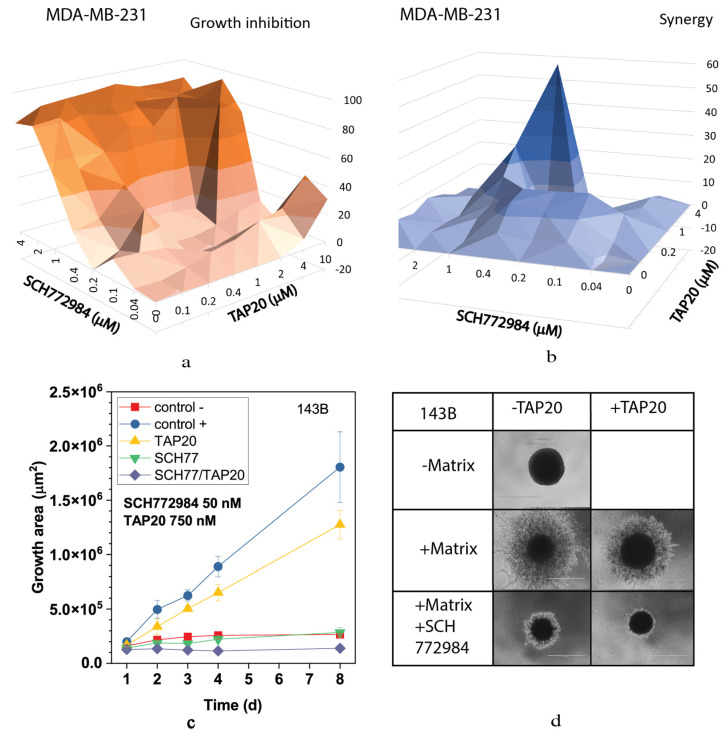
Synergy between TAP20 and SCH772984 and their effects on matrix-dependent invasion. For synergy analysis, MDA-MB-231 cells were grown in the presence of different combined concentrations of TAP20 and SCH772984 (n = 3). (**a**) Inhibition of growth by different combinations of TAP20 and SCH772984. (**b**) Synergy scores for the same combinations. For matrix-dependent invasion, (**c**,**d**) 143B osteosarcoma cells were grown to form spheroids before adding the extracellular matrix for invasion (n = 6). Invasion areas were measured by microscopy. Outgrowths with or without matrix are shown as controls. (**c**) Quantitative evaluation of outgrowths. (**d**) Examples of the effects SCH772894 and TAP20 on invasion. Outgrowths with or without matrix are shown as controls.

**Table 1 metabolites-13-01064-t001:** Compounds used in this study.

Compound	Target	IC_50_
Agerafenib	BRAF^V600E^	14 nM
AMG PERK44	PERK	6 nM
AZD3965	MCT1	1.6 nM
Bay876	GLUT1	2 nM
Bortezomib	20S proteasome	0.6 nM
Bumetanide	NKCC1	680 nM
CB-839	Glutaminase 1	25 nM
CB-5083	p97AAA ATPase	11 nM
Cetuximab	EGFR	0.2 nM
Danusertib	Aurora-kinases	13–79 nM
GSK2837808A	LDH-A	2.6 nM
GZD824	Bcr-Abl	0.32 nM
Flavopiridol	CDK (non-specific)	6–300 nM
5-Fluorouracil	Thymidylate synthase	Irr ^1^
Halofuginone	Prolyl-tRNA synthetase	18 nM
NCT-503	PHGDH	2500 nM
PLX8394	BRAF/BRAF^V600E^	14 nM/5 nM
Rapamycin	FKBP12	0.1 nM
SCH772984	ERK1/2	4 nM/1 nM
Seliciclib	CDK (non-specific)	200–800 nM
Selumetinib	MEK1/2	14 nM
Thapsigargin	SERCA	0.4 nM
THZ-1	CDK7	Irr ^1^
V-9302	SNAT2/LAT1	n.d.
YH16899	KRS-67LR interaction	8600 nM

^1^ irr: irreversible inhibitor. n.d.: not detected.

**Table 2 metabolites-13-01064-t002:** Sensitivity of cell lines to compounds used in this study. Inhibition of growth is shown as IC_50_ (μM). n.i. = no inhibition, n.d. = not determined. For each cell line n = 12–18 independent biological replicates were analyzed.

Compound	MDA-MB-231	MDA-MB-468	HPAFII	Panc02.03	SKOV3	OVCAR3
Agerafenib	4.3	4.3	1.6	3.7	4.7	6.5
AMG PERK44	>10	>10	9	>10	>10	>10
AZD3965	>10	>10	>10	>10	>10	>10
Bay876	10	10	0.25	0.15	1.4	0.16
Bortezomib	0.002	0.003	0.003	0.004	0.004	0.002
Bumetanide	n.i.	n.d.	n.i.	n.d.	n.i.	n.d.
CB-839	10	n.d.	>10	n.d.	>10	n.d.
CB-5083	0.76	0.36	0.19	0.12	1.2	0.36
Cetuximab	>10	n.d.	>10	n.d.	>10	n.d.
Danusertib	<0.1	<0.1	0.8	0.36	2.2	<0.1
GSK2837808A	>10	n.d.	>10	n.d.	>10	n.d.
GZD824	1	1.3	<0.1	0.31	1.6	1.8
Flavopiridol	0.1	0.04	<0.1	4	0.22	1.7
5-Fluorouracil	>10	>10	>10	8.9	>10	1.7
Halofuginone	0.032	0.054	0.038	0.057	0.019	0.058
NCT-503	>10	>10	>10	>10	>10	>10
PLX8394	10	10	4.5	10	6	3.6
Rapamycin	>10	n.d.	>10	n.d.	>10	n.d.
SCH772984	1	1	0.45	0.4	8.4	5
Seliciclib	>10	>10	9.5	6.3	>10	7.4
Selumetinib	10	6.3	<0.1	0.88	10	1.4
Thapsigargin	<0.1	0.005	<0.1	0.004	0.52	<0.003
V-9302	1.1	1.1	1.1	2	2.6	1.1
YH16899	>10	n.d.	>10	n.d.	>10	n.d

**Table 3 metabolites-13-01064-t003:** Compound concentrations used to detect synergy with TAP20. All concentrations are presented as μM.

**Compound**	**MDA-MB-231**	**MDA-MB-468**	**HPAFII**	**Panc02.03**	**SKOV3**	**OVCAR3**
Agerafenib	3	3	1	0.3	3	3
AMG PERK44	10	10	10	10	10	10
AZD3965	10	10	10	10	10	3
Bay876	10	3	0.1	0.03	1	0.1
Bortezomib	0.001	0.001	0.001	0.001	0.001	0.001
Bumetanide	10	10	10	10	10	10
CB-839	10	10	10	10	10	10
CB-5083	1	0.3	0.1	0.1	1	0.3
Cetuximab	0.033	0.033	0.033	0.033	0.033	0.033
Danusertib	0.01	0.03	0.3	0.1	1	0.01
GSK2837808A	10	10	10	10	10	10
GZD824	0.3	0.3	0.01	1	0.1	0.1
Flavopiridol	0.1	0.03	0.03	3	0.1	1
5-Fluorouracil	10	3	3	3	10	1
Halofuginone	0.01	0.01	0.01	0.01	0.01	0.01
NCT-503	10	10	10	10	10	10
PLX8394	10	3	3	3	3	1
Rapamycin	10	10	10	10	10	10
SCH772984	1	0.3	0.3	0.1	3	0.1
Seliciclib	10	3	3	3	10	3
Selumetinib	10	3	0.03	0.3	10	0.3
Thapsigargin	0.01	0.001	0.003	0.001	0.1	0.001
V-9302	1	1	1	1	1	1
YH16899	10	10	10	10	10	10

**Table 4 metabolites-13-01064-t004:** Synergy between TAP20 and experimental compounds. Coefficient of drug interaction (CDI) values are shown. Antagonism (>1, red), additive (1, yellow), synergistic (<1, green). Bold values are statistically significant (*p* < 0.05). For each combination n = 18 replicates were analyzed, except MDA-MB-231/THZ-1 (n = 6) and SKOV3/THZ-1 (n = 12) (n.d.: not detected).

Compound	MDA-MB-231	MDA-MB-468	HPAFII	Panc02.03	SKOV3	OVCAR3
Agerafenib	1.21	0.96	**0.97**	1.07	**1.49**	**1.57**
AMG PERK44	0.88	0.94	0.66	0.97	1.07	1.3
AZD3965	0.77	0.79	0.76	1.05	0.96	0.84
Bay876	0.97	0.63	0.73	0.73	0.87	0.72
Bortezomib	0.79	**0.82**	0.62	**0.83**	0.89	0.91
Bumetanide	0.83	0.81	0.86	0.99	0.97	0.93
CB-839	0.86	0.9	0.89	1.03	0.97	0.97
CB-5083	0.85	0.89	**0.51**	0.96	1.17	0.81
Cetuximab	1.11	0.77	1.06	0.97	0.99	0.91
Danusertib	0.85	0.65	**0.23**	**0.67**	0.65	1.53
GSK2837808A	0.85	0.87	0.78	1.03	0.99	0.99
GZD824	0.82	0.86	0.87	0.91	1.5	1.1
Flavopiridol	**0.24**	0.96	**0.4**	1.06	0.8	0.56
5-Fluorouracil	0.7	0.95	0.71	0.87	0.94	0.51
Halofuginone	**0.78**	0.88	**0.53**	**0.82**	0.73	0.93
NCT-503	**0.68**	0.88	0.74	**0.86**	0.95	1.11
PLX8394	0.91	0.6	0.88	0.88	1.16	0.97
Rapamycin	1.29	1.05	1.56	0.97	1.57	2.38
SCH772984	**0.23**	1.4	**0.29**	0.86	**0.53**	0.88
Seliciclib	**0.47**	0.85	**0.51**	0.96	0.87	0.88
Selumetinib	**0.3**	0.63	1.21	**0.7**	0.71	0.63
Thapsigargin	0.87	0.95	**0.57**	1.01	0.69	0.87
THZ-1	0.71	n.d.	**0.12**	n.d.	0.4	n.d.
V-9302	**0.63**	0.96	**0.44**	**0.84**	0.93	0.77
YH16899	**0.9**	**0.75**	**0.55**	0.94	0.93	0.76

## Data Availability

The data presented in this study are openly available in: Bröer, Stefan; Rahimi, Farid; Gauthier-Coles, Gregory; Broer, Angelika (2023), “Inhibition of GCN2 Reveals Synergy with Cell-Cycle Regulation and Proteostasis”, Mendeley Data, V1, doi: 10.17632/nhxphwtktr.1.

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
