# Peer review of "Inhibition of GCN2 Reveals Synergy with Cell-Cycle Regulation and Proteostasis"

_metabolites, 2023, doi:10.3390/metabo13101064_

Round 1

Reviewer 1 Report

The study is based on screening a combination of drugs to identify biologically active compounds which synergize with the GCN2 inhibitor TAP20 in 6 cancer cell lines. The authors further characterized the strongly synergistic combinations using cell viability and matrix-dependent invasion assays.

The article lacks experimentally validation. The authors must include additional experimentation to validate and enrich the present work.

Points that need to be validated:

1.      Please clarify the rationale for choosing the drugs included in the study with references.

2.      Provide the quantitation plot with significance for figure 1.

3.      Show the effect of TAP20 as an inhibitor for the panels of p-GCN2/GCN2, p-eIF2α/eIF2α and ATF4 in the same set of cells treated for different time points and increased dosage of the drug. Show the quantification bar with significance.

4.      The authors have shown the synergy study as a measure of cell death and matrix-dependent invasion upon different drug combination with TAP20 only in one breast cancer cell line. This result should be repeated in at least another 3 different cell lines for the validation.

5.      What is the effect of the treatments either TAP20 alone or with cell cycle inhibitors in combination for the cell cycle arrest? Analyze it using flow cytometry.

6.      The authors should perform apoptosis assay Annexin V/PI by flow cytometry to validate the cell death in single vs combination treatment.

7.       Did the authors tried to perform any in vivo studies to understand the synergistic effect in xenograft models for validation, at least for a single drug among the list either alone or in combination with TAP20.

8.      If not, the authors should try to perform at the least some cell viability experiments with TAP20 alone or in combination in primary cells cultured from patients samples.

9.      Discuss in details the clinical importance of the study. Are the drugs used already involved in some clinical trials in solid cancer?

Minor editing

Author Response

A manuscript with markup visible has been uploaded for reference purposes.

Reviewer 1:

The study is based on screening a combination of drugs to identify biologically active compounds which synergize with the GCN2 inhibitor TAP20 in 6 cancer cell lines. The authors further characterized the strongly synergistic combinations using cell viability and matrix-dependent invasion assays.

The article lacks experimentally validation. The authors must include additional experimentation to validate and enrich the present work.

Response: The authors disagree with the overall statement. The main purpose of the study was to screen for synergy between GCN2 inhibitor TAP20 and a selection of experimental and approved drugs. The identified hits were validated by detailed synergy analysis using a matrix of concentration combinations. We admit that this study is descriptive and that mechanistic explanations warrant further experimentation.

Points that need to be validated:

  1. Please clarify the rationale for choosing the drugs included in the study with references.

Response: This information has been added to the manuscript.

  1. Provide the quantitation plot with significance for figure 1.

Response: The quantification has been added as a supplementary figure.

  1. Show the effect of TAP20 as an inhibitor for the panels of p-GCN2/GCN2, p-eIF2α/eIF2α and ATF4 in the same set of cells treated for different time points and increased dosage of the drug. Show the quantification bar with significance.

Response: Due to the cell culturing required, these experiments cannot be performed in the 10-day response period for a major revision.

  1. The authors have shown the synergy study as a measure of cell death and matrix-dependent invasion upon different drug combination with TAP20 only in one breast cancer cell line. This result should be repeated in at least another 3 different cell lines for the validation.

Response: We have used an osteosarcoma cell line for matrix-dependent invasion assays. As outlined in the manuscript this cell line was chosen because of its invasion properties. The cell lines tested for synergy did not show sufficient invasive capacity to be analysed in this way.

  1. What is the effect of the treatments either TAP20 alone or with cell cycle inhibitors in combination for the cell cycle arrest? Analyze it using flow cytometry.

Response: The primary purpose of this study was to identify synergy between TAP20 and selected experimental and approved cancer drugs. As such, the study is descriptive and explorative. Analysing the mechanism of inhibition goes beyond this study, because each combination may have a different mechanism of synergy.

  1. The authors should perform apoptosis assay Annexin V/PI by flow cytometry to validate the cell death in single vs combination treatment.

Response: The primary purpose of this study was to identify synergy between TAP20 and selected experimental and approved cancer drugs. As such, the study is descriptive and explorative. Analysing the mechanism of inhibition goes beyond this study, because each combination may have a different mechanism of synergy.

  1. Did the authors tried to perform any in vivo studies to understand the synergistic effect in xenograft models for validation, at least for a single drug among the list either alone or in combination with TAP20.

Response: This is an unreasonable demand. Such studies would take more than a year to complete and use large numbers of animals.

  1. If not, the authors should try to perform at the least some cell viability experiments with TAP20 alone or in combination in primary cells cultured from patients samples.

Response: While informative, these studies cannot be completed in the 10-day time-frame of a major revision. Setting up these experimental systems would take many months.

  1. Discuss in details the clinical importance of the study. Are the drugs used already involved in some clinical trials in solid cancer?

Response: TAP20 is an experimental preclinical compound. CDK inhibitors have been tested in clinical trials but toxicity has been a problem. We have expanded the discussion along these lines. We have also added to the discussion where combination trials have been made with compounds used in this study.

Reviewer 2 Report

Review on the manuscript of Gauthier-Coles, G. et al.: “Inhibition of GCN2 reveals synergy with cell-cycle regulation and proteostasis”.

In this manuscript, authors explored the sensivity of cancer cell lines to the GCN2 inhibitor, TAP20, combined with 25 compounds targeted to distinct cellular pathways/targets. Authors observed that part of these combinations were synergistic, suggesting that a combined therapy of GCN2 inhibitors with molecules targeting other cellular pathways could be beneficial for some types of cancer

The manuscript is clear and precise on the questions that authors proposed to answer. Thus, the issues that arise to me are listed below, so, I hope the authors find the following comments and suggestions useful.

1 – The ATF4 expression in Pac 02.03 cells does not directly correlates with the TAP20 concentration (increasing the concentration of TAP20 does not proportionally decreases ATF4 expression) Do Authors have an explanation for that?

2 – In the unprocessed growth data shown in supplementary Fig. S2, the results for TAP20 alone are not included. I recommend authors to include the TAP20 bar in each graph to make the results clearer.

3 – As illustrated in Table 4, most of the synergistic effects between TAP20 and the tested compounds were observed in the HPAFII cell line. Do Authors have an explanation for that?

4 – I encourage Authors to rearrange the data included in the Graphs shown in figures 2-7. Instead of making groups for treatment condition (for each condition, the different days are shown together), I would recommend Authors to make groups for the day. For example, by making a group for day 1, another group for day 2, …, all treatment conditions are close together. This way, for each day, it is easier to understand the effect of each treatment compared to the other conditions.

5 - In the “Funding” and “Conflicts of Interest” sections, I recommend Authors to indicate the reference of the research contract provided by Merck KgAa.

Author Response

A manuscript with markup visible has been uploaded for reference purposes.

Reviewer 2:

Review on the manuscript of Gauthier-Coles, G. et al.: “Inhibition of GCN2 reveals synergy with cell-cycle regulation and proteostasis”.

In this manuscript, authors explored the sensivity of cancer cell lines to the GCN2 inhibitor, TAP20, combined with 25 compounds targeted to distinct cellular pathways/targets. Authors observed that part of these combinations were synergistic, suggesting that a combined therapy of GCN2 inhibitors with molecules targeting other cellular pathways could be beneficial for some types of cancer

The manuscript is clear and precise on the questions that authors proposed to answer. Thus, the issues that arise to me are listed below, so, I hope the authors find the following comments and suggestions useful.

1 – The ATF4 expression in Pac 02.03 cells does not directly correlates with the TAP20 concentration (increasing the concentration of TAP20 does not proportionally decreases ATF4 expression) Do Authors have an explanation for that?

Response, we have added to the discussion: The only exception was Panc 02.03 cells, where ATF4 expression could not be suppressed by addition of TAP20. This suggests that ATF4 expression may have been upregulated by a GCN2 independent mechanism, such as PERK. Consistently, we found limited synergy in our screen with this cell line.

2 – In the unprocessed growth data shown in supplementary Fig. S2, the results for TAP20 alone are not included. I recommend authors to include the TAP20 bar in each graph to make the results clearer.

Response: The reviewer may have overlooked this. The red dashed line Fig. S2 is the TAP20 result.

3 – As illustrated in Table 4, most of the synergistic effects between TAP20 and the tested compounds were observed in the HPAFII cell line. Do Authors have an explanation for that?

Response, we have added to the discussion: In this screen we found that synergy is strongly cell-line dependent. A recent systematic survey of drug interactions by Jaaks et al. found that only 5.2% of >100,000 combinations were synergistic (ref). Intriguingly, the highest rate of synergy was found in pancreas cancer. Here we also identified HPAF-II cells as particularly sensitive to combination treatment. Due to the exocrine origin of pancreatic adenocarcinomas protein synthesis is likely to be high and therefore HPAF-II cells might be particularly vulnerable to amino acid limitation. The GCN2-ATF4 pathway as a stress-induced response is more likely to result in synergistic relationships, potentially explaining our higher rate of synergism of 11% in the overall panel. The study by Jaaks et al. (ref) also reported high synergy rates with cell cycle inhibitors.

4 – I encourage Authors to rearrange the data included in the Graphs shown in figures 2-7. Instead of making groups for treatment condition (for each condition, the different days are shown together), I would recommend Authors to make groups for the day. For example, by making a group for day 1, another group for day 2, …, all treatment conditions are close together. This way, for each day, it is easier to understand the effect of each treatment compared to the other conditions.

Response: The authors appreciate the suggestion, all figures have been updated accordingly.

5 - In the “Funding” and “Conflicts of Interest” sections, I recommend Authors to indicate the reference of the research contract provided by Merck KgAa.

Response: Curiously, the research contract did not have a reference number.

Reviewer 3 Report

In this manuscript, the authors describe a study focused on the integrated stress response, particularly one of its branches involving GCN2, which senses amino acid insufficiency and plays a role in maintaining amino acid homeostasis. The research explores the potential of GCN2 as a cancer therapeutic target when combined with the GCN2 inhibitor TAP20. The study involves screening a panel of 25 compounds for their synergistic effects with TAP20 on cell growth in multiple cancer cell lines.

The manuscript should be accepted for publication in the journal after major revision, on the following section:

-        Line 7: Remove correspondence reported twice.

-        The authors should add GCN2 among the keywords.

-        In the introduction, the authors should emphasize why understanding the ISR pathway and the role of GCN2 in cancer research is significant. Clearly state the purpose of the study and the potential implications of the research.

-        To my best knowledge, GCN2 activation affects cell-cycle regulation indirectly through its impact on global protein synthesis and the levels of key proteins, including cyclins and CDKs. When GCN2 is activated in response to stress, it can lead to reduced protein synthesis, decreased cyclin levels, and cell-cycle arrest, primarily at the G1 phase. Additionally, GCN2 activation can also lead to the activation of p53, a tumor suppressor protein. p53 has been shown to regulate CDK activity and cell-cycle progression in response to stress and DNA damage. Activated p53 can induce cell-cycle arrest by inhibiting CDK-cyclin complexes and promoting the expression of CDK inhibitors.

The authors should elaborate on this aspect in the manuscript, referencing some very recent papers that must be added to the references:

·       Cancers (Basel). 2021 Aug 30;13(17):4389. doi: 10.3390/cancers13174389.

-        Lines 356-357: The sentence “Combination of drugs is an important strategy to overcome drug resistance and to inhibit processes that require significant pathway flexibility such as metastasis” needs to be referenced. This adds credibility to the information and allows readers to explore the sources for more details. I suggest adding this reference:

·       Curr Med Chem. 2023;30(7):776-782. doi: 10.2174/0929867329666220729152741.

·       Cell Rep. 2023 Jun 1;42(6):112581. doi: 10.1016/j.celrep.2023.112581.

-        Lines 398-400: I suggest adding a reference along with your reference 44, related to the following paper that clearly explains the Regulation of mTORC1 through the TSC complex:

·       Mol Cancer 22, 138 (2023). https://doi.org/10.1186/s12943-023-01827-6

 Minor editing of English language required

Author Response

A manuscript with markup visible has been uploaded for reference purposes.

Reviewer 3:

In this manuscript, the authors describe a study focused on the integrated stress response, particularly one of its branches involving GCN2, which senses amino acid insufficiency and plays a role in maintaining amino acid homeostasis. The research explores the potential of GCN2 as a cancer therapeutic target when combined with the GCN2 inhibitor TAP20. The study involves screening a panel of 25 compounds for their synergistic effects with TAP20 on cell growth in multiple cancer cell lines.

The manuscript should be accepted for publication in the journal after major revision, on the following section:

-Line 7: Remove correspondence reported twice.

Response: removed as suggested.

-The authors should add GCN2 among the keywords.

Response: The authors have been advised that terms appearing in the title do not need to be repeated in the key words as both are indexed.

-In the introduction, the authors should emphasize why understanding the ISR pathway and the role of GCN2 in cancer research is significant. Clearly state the purpose of the study and the potential implications of the research.

Response, we have added to the introduction: GCN2 is activated in cancers that experience stress due to nutrient depletion in the tumor environment, elevated protein biosynthesis and/or limiting vascularisation (ref). Activation of GCN2 allows tumor cells to adapt to changes in their microenvironment and promotes quiescence thereby causing therapy resistance (ref). Thus, combining conventional cancer therapy with concomitant inhibition of GCN2 could overcome therapy resistance and improve treatment outcomes.

  • To my best knowledge, GCN2 activation affects cell-cycle regulation indirectly through its impact on global protein synthesis and the levels of key proteins, including cyclins and CDKs. When GCN2 is activated in response to stress, it can lead to reduced protein synthesis, decreased cyclin levels, and cell-cycle arrest, primarily at the G1 phase. Additionally, GCN2 activation can also lead to the activation of p53, a tumor suppressor protein. p53 has been shown to regulate CDK activity and cell-cycle progression in response to stress and DNA damage. Activated p53 can induce cell-cycle arrest by inhibiting CDK-cyclin complexes and promoting the expression of CDK inhibitors.

The authors should elaborate on this aspect in the manuscript, referencing some very recent papers that must be added to the references:

  • Cancers (Basel). 2021 Aug 30;13(17):4389. doi: 10.3390/cancers13174389.

Response: The authors appreciate the insightful comment. We have added to the discussion: Phosphorylation of eIF2α by GCN2 or PERK has been shown to inhibit translation of cyclin D1 thus leading to cell-cycle arrest at the G1 phase (ref). In addition, amino acid deprivation stops cell-cycle progression by stabilizing the mRNA of endogenous cyclin-dependent kinase inhibitors p22 and p27 (ref). Starved cells withdraw from the cell cycle by a mechanism involving transcription factor p53 (ref). This is mediated by stimulation of the cyclin-dependent kinase 1 inhibitor p21 (ref). Thus, multiple mechanisms connect amino acid homeostasis with cell cycle control.

and

Clinical trials with CDK inhibitors such as flavopiridol have shown limited efficacy even in combination with other chemotherapeutics (ref). This could be due to activation of GCN2, thereby causing cells to become quiescent. The non-specific CDK inhibitor seliciclib showed significant toxicity in clinical studies at doses that correspond to effective concentrations in preclinical models (ref).

-Lines 356-357: The sentence “Combination of drugs is an important strategy to overcome drug resistance and to inhibit processes that require significant pathway flexibility such as metastasis” needs to be referenced. This adds credibility to the information and allows readers to explore the sources for more details. I suggest adding this reference:

  • Curr Med Chem. 2023;30(7):776-782. doi: 10.2174/0929867329666220729152741.

Cell Rep. 2023 Jun 1;42(6):112581. doi: 10.1016/j.celrep.2023.112581.

-Lines 398-400: I suggest adding a reference along with your reference 44, related to the following paper that clearly explains the Regulation of mTORC1 through the TSC complex:

  • Mol Cancer 22, 138 (2023). https://doi.org/10.1186/s12943-023-01827-6

Response: We have added these references as suggested and expanded the discussion.

Round 2

Reviewer 1 Report

The authors modified the MS to some extent but I have expected they should perform few doable experiments that could have enriched the MS. 

Minor

Reviewer 2 Report

Second review on the manuscript of Gauthier-Coles, G. et al.: “Inhibition of GCN2 reveals synergy with cell-cycle regulation and proteostasis”.

In this manuscript, authors explored the sensivity of cancer cell lines to the GCN2 inhibitor, TAP20, combined with 25 compounds targeted to distinct cellular pathways/targets. Authors observed that part of these combinations were synergistic, suggesting that a combined therapy of GCN2 inhibitors with molecules targeting other cellular pathways could be beneficial for some types of cancer

This corresponds to a second version of the manuscript after peer-review. I consider that Authors have made a great job in improving the manuscript according with the Reviewer’s suggestions.